# Analysis and Design of a Non-Magnetic Bulk CMOS Passive Circulator Using 25% Duty-Cycle Clock

**DOI:** 10.3390/mi14010033

**Published:** 2022-12-23

**Authors:** Jian Gao, Xinghua Wang, Fang Han, Jiayue Wan, Wei Gu

**Affiliations:** 1School of Integrated Circuits and Electronics, Beijing Institute of Technology, Beijing 100081, China; 2School of Information and Electronics, Beijing Institute of Technology, Beijing 100081, China

**Keywords:** CMOS, circulator, 25% duty cycle, full-duplex, gyrator

## Abstract

A circulator, which is a non-reciprocal device, is widely used in full-duplex systems, future communication and sensing networks, and quantum computing, and it is difficult to implement a passive topology on a chip. Based on switch-based spatio-temporal conductivity modulation, in this study, we design and implement a non-magnetic on-chip passive circulator operating at the Ku band in a 90-nm bulk CMOS technology using a 25% duty-cycle I/Q clock signal. With the virtue of the four-phase non-overlapping clock signal, the proposed circulator achieves a 3.9 dB transmitter (TX)-to-antenna (ANT) and a 4.0 dB ANT-to-receiver (RX) insertion loss with a 1-dB bandwidth of 2.7 GHz (21.4%). The TX-to-RX isolation is better than 17.2 dB, and the TX-to-ANT IIP3 and ANT-to-RX IIP3 are 19.7 dBm and 20.0 dBm, respectively, while occupying a die area of 1.55 mm × 1.15 mm. Although low-cost bulk CMOS technology is used, competitive isolation, linearity performance, and isolation bandwidth are achieved in the proposed design.

## 1. Introduction

Full-duplex (FD) communication, considered as one of the most important technologies in 5G as well as future communication and sensing systems, has been widely studied in recent years [1,2,3,4,5,6,7], as it allows simultaneous transmission and reception at the same time. Compared to Frequency Division Duplexing (FDD) and Time Division Duplexing (TDD) systems that use different frequency channels or time slots for transmission and reception, FD may increase the transmission speed by up to two times and save the frequency bands. However, the most significant obstacle for FD to be commercially used is the large self-interference (SI) from transmitter to receiver [3]. Therefore, a number of self-interference cancellation (SIC) technologies have been proposed in terms of the antenna interface, radio frequency (RF), and analog and digital domains [8,9,10,11,12,13,14,15,16,17,18] in order to mitigate this challenge.

At the antenna interface, SIC can be performed by separated antennas [7], which occupy a large area and are not suitable for phased-array or multiple input, multiple output (MIMO) systems. In order to reduce the area, a shared antenna is suggested. Typically, an electrical-balance duplexer (EBD) [19,20] is utilized to suppress SI. However, the EBD, which is a reciprocal three-port device, exhibits an inherent 3-dB loss, which is not acceptable in low-loss systems nowadays. Moreover, another candidate is a circulator, which is a non-reciprocal three-port device. Traditionally, a circulator is implemented using ferrite materials, which are bulky, expensive, and incompatible with the integrated circuit (IC) fabrication process. On-chip active circulators [21,22,23,24,25] using transistors suffer from poor noise and linearity performance and a limited dynamic range [26]. Therefore, a non-magnetic non-reciprocal passive on-chip circulator is expected. Nevertheless, the implementation of non-magnetic non-reciprocity is a great challenge, which has been widely studied in the past few years [27,28,29,30,31,32,33,34,35].

An on-chip CMOS non-magnetic passive circulator was first demonstrated by Prof. Krishnaswamy’s group using an N-path filter [32,36]. Later, the first on-chip non-magnetic passive circulator based on switch-based spatio-temporal conductivity modulation [33] was proposed [37], which consumes more area. Compared to the circulator using the N-path filter, a higher operating frequency and lower clock frequency were achieved. It is also utilized in quantum computing applications [38,39] at cryogenic temperatures. A summary of the previous literature is shown in Table 1.

Nowadays, the circulator based on the N-path filter is attempted to be used in complete full-duplex transceivers [36] due to its small form factor and lower insertion loss. However, the circulator based on the N-path filter limits the transceivers to operate at lower frequencies, which is not suitable for millimeter-wave wireless systems. Traditionally, a 50% duty-cycle clock signal is applied on the circulator based on switch-based spatio-temporal conductivity modulation. However, using a 25% duty-cycle clock signal can potentially provide a higher conversion gain and lower crosstalk compared with using a 50% duty-cycle clock signal. Moreover, the design of two local oscillator (LO) signals at millimeter-wave frequencies with different duty cycles is a great challenge.

According to the previous discussion, a passive circulator operating at 10–20 GHz remains to be studied. In this frequency band, Ku-band satellite communication is located. Traditionally, the simultaneous transmission and reception of a satellite communication system is implemented by utilizing different frequency channels. However, with the increasing data volume and the emerging future communication methods, full-duplex satellite communication becomes a promising means to potentially double the throughput and save the frequency bands. Therefore, an on-chip passive circulator for full-duplex satellite communication is an emerging research topic in the near future.

To address the research gap and potentially use the 25%-duty-cycle clock signal to implement higher conversion gains, this paper describes the analysis, design, and implementation of a non-magnetic passive circulator operating at the Ku band based on switch-based spatio-temporal conductivity modulation in a 90-nm bulk CMOS technology. Section 2 begins with an illustration of the working state of the proposed circulator based on time-domain and frequency-domain analysis. A novel system-level analysis method is also presented in this section. Then, the implementation details of the proposed circulator are provided in Section 3. Section 4 and Section 5 presents the measurement results and the limitation of the study and recommendations, and the paper concludes in Section 6.

## 2. Analysis

A gyrator [46], considered as a passive, non-reciprocal, and two-port device, was proposed in 1948, which exhibits a phase difference of 180° between the forward and reverse directions and is a good candidate for passive non-reciprocal element design, e.g., circulators. The gyrator using a 25% duty-cycle clock signal embedded in the proposed circulator is shown in Figure 1. In order to clarify the working state of the gyrator, time-domain analysis, frequency-domain analysis, and the novel system-level analysis are presented in this section.

### 2.1. Time-Domain Analysis

In order to verify the working state of the gyrator using a 25% duty-cycle clock signal, the phase shift of different modulation periods both in forward and reverse directions is illustrated, as shown in Table 2 and Table 3.

From Table 2 and Table 3, it is found that the gyrator using the 25% duty-cycle clock signal exhibits a 90° phase shift in the forward direction and −90° in the reverse direction, which implements a phase difference of 180°.

### 2.2. Frequency-Domain Analysis

In this section, the frequency-domain analysis of the gyrator embedded in the proposed circulator is explained. Due to the double-balanced differential implementation of the switches, the final derivation can be multiplied by 2 directly. The Fourier series of the 25% duty-cycle clock signal is as follows:(1)$$LO\left(t\right)=\frac{2}{\pi }\sum_{n=1}^{\infty }\frac{\sin\frac{\left(2n-1\right)\pi }{4}}{2n-1}\left(e^{j\left(2n-1\right)\omega_{m}t}+e^{-j\left(2n-1\right)\omega_{m}t}\right)$$

For an input signal $v_{1}^{+}\left(t\right)=e^{j\omega t}$, the output signal of the gyrator at the right-hand side can be derived as
(2)$$\begin{array}{l} v_{2}^{-}\left(t\right)=v_{1}^{+}\left(t-\frac{T_{m}}{4}\right)\times LO\left(t-\frac{T_{m}}{4}\right)\times LO\left(t-\frac{T_{m}}{4}\right)\\ \;\;\;\;\;\;=\frac{8}{\pi^{2}}e^{-j\omega \frac{T_{m}}{4}}e^{j\omega t}\sum_{n=1}^{\infty }\frac{\sin^{2}\left(\frac{2n-1}{4}\right)}{{\left(2n-1\right)}^{2}}\\ \;\;\;\;\;\;=\frac{1}{2}\cdot \frac{8}{\pi^{2}}e^{-j\omega \frac{T_{m}}{4}}e^{j\omega t}\sum_{n=1}^{\infty }\frac{1}{{\left(2n-1\right)}^{2}}\\ \;\;\;\;\;\;=\frac{1}{2}e^{-j\omega \frac{T_{m}}{4}}e^{j\omega t} \end{array}$$

Therefore, the total output signal mixed by the four-phase clock signal can be calculated as
(3)$$\begin{array}{l} v_{2}^{-}{\left(t\right)}_{total}=e^{-j\omega \frac{T_{m}}{4}}e^{j\omega t}\\ \;\;\;\;\;\;\;\;\;\;\;\;=e^{-j\omega \frac{T_{m}}{4}}v_{1}^{+}\left(t\right) \end{array}$$

Similar derivation can be performed for the reverse direction and the final result is as follows:(4)$$\begin{array}{l} v_{1}^{-}{\left(t\right)}_{total}=-e^{-j\omega \frac{T_{m}}{4}}e^{j\omega t}\\ \;\;\;\;\;\;\;\;\;\;\;\;=-e^{-j\omega \frac{T_{m}}{4}}v_{2}^{+}\left(t\right) \end{array}$$

Equations (3) and (4) show that the ideal gyrator using a 25% duty-cycle clock signal is lossless and appears externally as linear time invariant (LTI), which has the same results compared with the gyrator using a 50% duty-cycle clock signal [34].

### 2.3. System-Level Analysis

In order to provide a more straightforward and accurate analysis of the gyrator’s performance, a system-level analysis method is proposed in this section.

Starting from Equation (1), when the LO signal is mixed with an input signal expressed as $v_{1}^{+}\left(t\right)=e^{j\omega t}$, the output signal *h*(*t*) is shown as follows:(5)$$\begin{array}{l} h\left(t\right)=LO\left(t\right)\times v_{1}^{+}\left(t\right)\\ \;\;\;\;=\frac{2}{\pi }\sum_{n=1}^{\infty }\frac{\sin\frac{\left(2n-1\right)\pi }{4}}{2n-1}\left(e^{j\left(\omega +\left(2n-1\right)\omega_{m}\right)t}+e^{j\left(\omega -\left(2n-1\right)\omega_{m}\right)t}\right) \end{array}$$

Now, if the output signal in Equation (5) is directly mixed again with the square wave in Equation (1), the output signal can be derived as
(6)$$\begin{array}{l} v_{2}^{-}\left(t\right)=LO\left(t\right)\times v_{1}^{+}\left(t\right)\times LO\left(t\right)\\ \;\;\;\;\;\;\;=\frac{2}{\pi }\sum_{n=1}^{\infty }\frac{\sin\frac{\left(2n-1\right)\pi }{4}}{2n-1}\left(e^{j\left(\omega +\left(2n-1\right)\omega_{m}\right)t}+e^{j\left(\omega -\left(2n-1\right)\omega_{m}\right)t}\right)\\ \;\;\;\;\;\;\;\times \frac{2}{\pi }\sum_{n=1}^{\infty }\frac{\sin\frac{\left(2n-1\right)\pi }{4}}{2n-1}\left(e^{j\left(2n-1\right)\omega_{m}t}+e^{-j\left(2n-1\right)\omega_{m}t}\right)\\ \;\;\;\;\;\;\;=\frac{1}{2}e^{j\omega t} \end{array}$$

Then, the total output signal mixed by the four-phase clock signal can be calculated as
(7)$$v_{2}^{-}{\left(t\right)}_{total}=v_{1}^{+}\left(t\right)$$

From Equations (5)–(7), it is found that the system illustrated above is independent with mixing products and is externally LTI and reciprocal as
(8)$$v_{2}^{-}\left(t-\tau \right)=LO\left(t-\tau \right)\times v_{1}^{+}\left(t-\tau \right)\times LO\left(t-\tau \right)$$
and
(9)$$\begin{array}{l} av_{2,a}^{-}\left(t-\tau \right)+bv_{2,b}^{-}\left(t-\tau \right)\\ =LO\left(t-\tau \right)\times \left(av_{1,a}^{+}\left(t-\tau \right)+bv_{1,b}^{+}\left(t-\tau \right)\right)\times LO\left(t-\tau \right) \end{array}$$

The system derived in Equations (5)–(9) can be illustrated as in Figure 2a,b. From another point of view, the system with time delay *τ* in Figure 2b can be separated into two sub-systems in the forward and reverse direction if the time delay is performed by the time delay module internally, as shown in Figure 3a,b. Note that the time delay module should be non-reciprocal and the system implements a time delay of ±*τ* in the forward and reverse direction.

If the time delay module is implemented by a transmission line, it must be reciprocal in both directions. Therefore, in order to implement the system shown in Figure 3a,b using a transmission line, the relationship shown below is required:(10)h(t+τ)=h(t−τ)

Considering the frequency components in the internal node after mixing expressed as
(11)$$\omega \pm \left(2n-1\right)\omega_{m}$$
when *τ* = *T_m_*/*n* and the frequency of the input signal is the harmonics of *ω_m_*, the ratio *n* that satisfies Equation (10) is analyzed. Under such a circumstance, the frequency components in the internal node are the harmonics of *ω_m_*, and if the minimum frequency component meets the requirement of Equation (10), all the higher-order harmonics will be satisfied. Thus, the minimum frequency component is analyzed here.

When the frequency of the input signal is the even harmonics of *ω_m_*, the minimum frequency component in the internal node is *ω_m_*, i.e.,
(12)$$\begin{array}{c} 2\tau =T_{m}\\ n=2 \end{array}$$

In this case, a reciprocal phase shift of *T_m_*/2 in the forward and reverse direction is implemented.

When the frequency of the input signal is the odd harmonics of *ω_m_*, the minimum frequency component is 2*ω_m_*, i.e.,
(13)$$\begin{array}{c} 2\tau =\frac{T_{m}}{2}\\ n=4 \end{array}$$

In this case, a non-reciprocal phase shift of ±*T_m_*/4 in the forward and reverse direction is implemented and the function of a gyrator is realized, which is used in the designed circulator.

## 3. Implementation

A comprehensive research framework is shown in Figure 4. The implementation details are illustrated in this section and the design procedure of the artificial transmission line is followed according to [37].

### 3.1. Generation of 25% Duty-Cycle Clock Signal 

Conventionally, a 25% duty-cycle clock signal is generated using a divide-by-2 frequency divider and AND gates, as shown in Figure 5, which results in high power consumption and poor phase-noise performance. In order not to deteriorate the performance of the whole full-duplex communication transceiver and implement a high-frequency, high-performance 25% duty-cycle clock signal, the generator that outputs the 25% duty-cycle clock signal directly after the divide-by-2 divider is used in this design, as shown in Figure 6.

The sinusoidal input signal with a frequency of 2*f_LO_* is injected using the ground-signal-ground (GSG) pad and the single-to-differential conversion is performed on the chip. A four-stage inverter buffer with resistor feedback in the second and fourth stages is applied directly after the 25% duty-cycle quadrature output. In order to mitigate the influence of the long-distance layout routing of the clock signal to the switches, another two-stage inverter buffer with resistor feedback in the second stage is placed near to the switches in the layout design.

A 2*f_LO_* (i.e., 8.4 GHz) sinusoidal signal is used as the input of the generator. The clock signal generator consumes power of 47.3 mW, which includes 3.9 mW from the generator core and 43.4 mW from all the buffers. After the inverter buffer chain, a four-phase 25% duty-cycle square-wave clock signal, of which the rising time is approximately 1/12 of the modulation period, is generated.

### 3.2. Proposed Circulator

The passive on-chip non-reciprocal CMOS circulator is proposed in this section, as shown in Figure 7, embedded with the gyrator using a 25% duty-cycle clock signal. The operating frequency is designed to be 12.6 GHz with a 4.2 GHz modulation clock. In order to absorb the parasitic capacitance of the switches to the λ/8 transmission line at 12.6 GHz, a symmetric structure is utilized in this design.

Considering the fabrication technology limitations, bulk CMOS transistors without a thick gate are used as the mixer switches, of which the parasitic capacitance is absorbed in the gyrator symmetrically.

The λ/8 transmission line at 12.6 GHz between the TX/RX port and gyrator is implemented by the *LC* lumped element, and the λ/4 transmission line at 12.6 GHz between the TX/RX port and ANT port is implemented by a grounded coplanar waveguide (GCPW). Taking the size, insertion loss, and bandwidth into consideration, the λ/4 transmission line at 4.2 GHz in the gyrator is realized by a 5-stage *LC* π-type lumped element transmission line. The simulated Bragg frequency is approximately 42 GHz, which can tolerate up to seven-times mixing products and is limited by the operating frequency and process used. Recently, the *LC* π-type lumped element transmission line has been modified to obtain a lower insertion loss, such as a hybrid π-lattice structure [42], band-pass filter structure [43], all-pass filter structure [39], etc., which are also suitable in the proposed circulator. In addition, in order to enable single-ended measurements, baluns are attached to the proposed design.

## 4. Measurements

The proposed on-chip passive circulator is fabricated in a 90-nm bulk CMOS technology. The die micro-photograph of the chip is shown in Figure 8, and the total chip area is 1.55 mm × 1.15 mm with the TX and RX baluns and bonding pads excluded (ANT balun is difficult to de-embed, as shown in Figure 7). The measurement is performed by an RF probe with the die mounted on a PCB board. The power supply and ground are wire bonded to the board.

Four GSG pads are included on the chip for the probing test. Three of them are TX, ANT, and RX terminals and the other one is for clock signal input. The 8.4 GHz sinusoidal input clock signal is generated using a millimeter-wave signal generator and the single-to-differential conversion is implemented on the chip, as illustrated above. S-parameters are measured using a vector network analyzer with two ports probed at a time and the third port terminated with 50 Ω. On-chip calibration of the vector network analyzer is performed ahead of S-parameter measurement. The measured S-parameters of TX-to-ANT (port 1 and port 2), ANT-to-RX (port 2 and port 3) and TX-to-RX (port 1 and port 3) are shown in Figure 9a–c. The insertion loss (IL) is 3.9 dB from TX to ANT (S_21_) and 4.0 dB from ANT to RX (S_32_). The TX-to-RX isolation (S_31_) is >17.2 dB. The S-parameter results are concluded without any on-chip reconfiguration and ANT port impedance tuning, which leads to the reflection at the ANT port and influences the isolation performance. The bandwidth of 1-dB IL degradation is 2.7 GHz (21.4%), corresponding to an isolation of 17.2 to 21.7 dB.

The linearity of the proposed circulator is measured with a two-tone (12.6 GHz and 12.7 GHz) third-order intercept point (IP3) test. The fundamental output amplitude is extrapolated along a line of unity slope and the third-order intermodulation product (IM3) is extrapolated along a line with a slope of three, with the results exhibited in Figure 10a,b. It shows that an input third-order intercept point (IIP3) of 19.7 dBm from TX to ANT and 20.0 dBm from ANT to RX are achieved.

A performance summary and comparison is shown in Table 4. Compared to the prior work, this paper has implemented an on-chip non-reciprocal passive circulator using bulk CMOS technology operating at the Ku band. Thanks to the high-speed 25% duty-cycle clock signal generation circuit, the proposed circulator realizes competitive performance in isolation and linearity compared with the designs using SOI technology, which makes the cost lower. Although process limitations exist, the insertion loss from TX to ANT and from ANT to RX only deteriorates slightly compared to the design in CMOS SOI, which is significantly influenced by the process and operating frequency.

## 5. Limitations of the Study and Recommendations

Due to the lack of a qualified pre-amplifier, the followings are absent: noise figure, P1dB, and spurious tone measurement. However, P1dB and spurious tones can be reflected by IIP3 to some extent, and the noise figure should be slightly larger than the insertion loss in this passive circulator due to the phase noise of the clock signal.

From the current measurement results, it is recommended to use this circulator in a full-duplex system operating at around 10–20 GHz and requiring a low cost. Although the insertion loss is slightly larger than designs using CMOS SOI technology, other measurement results are competitive, and, moreover, the isolation bandwidth is slightly larger without any tuning method. The proposed design is a promising candidate for Ku-band full-duplex satellite communication.

## 6. Conclusions

This paper presents an on-chip non-reciprocal passive circulator using bulk CMOS technology with spatio-temporal conductivity modulation operating at the Ku band. Fabricated in a 90-nm bulk CMOS technology, the proposed 12.6-GHz circulator implements an insertion loss of 3.9 dB from TX to ANT and of 4.0 dB from ANT to RX, and with 1-dB degradation over 2.7 GHz, which corresponds to a bandwidth of 21.4%. A > 17.2 dB isolation is achieved, which ranges from 17.2 dB to 21.7 dB in the 1-dB degradation insertion loss bandwidth. Linearity performance is also competitive compared to the prior work using SOI technology without antenna impedance tuning. The proposed circulator consumes an area of 1.55 mm × 1.15 mm and is suitable for low-cost RFIC design.

## Figures and Tables

**Figure 1 micromachines-14-00033-f001:**
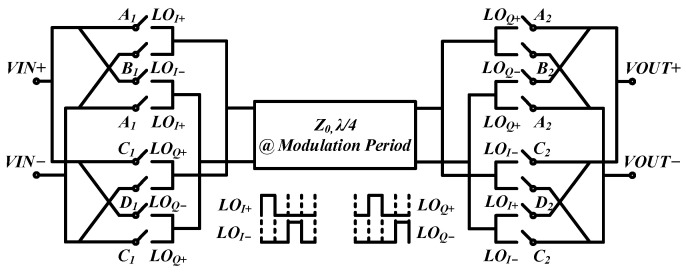
Gyrator using 25% duty-cycle clock signal.

**Figure 2 micromachines-14-00033-f002:**
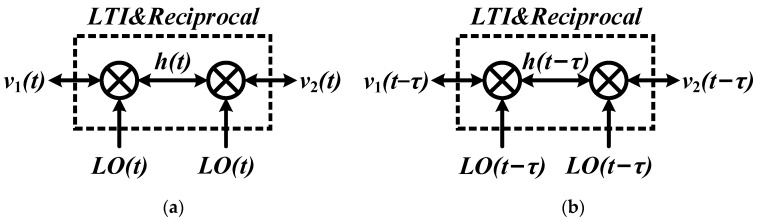
Doubly square-wave mixing system with LTI and reciprocal property: (**a**) original case; (**b**) with time delay *τ*.

**Figure 3 micromachines-14-00033-f003:**
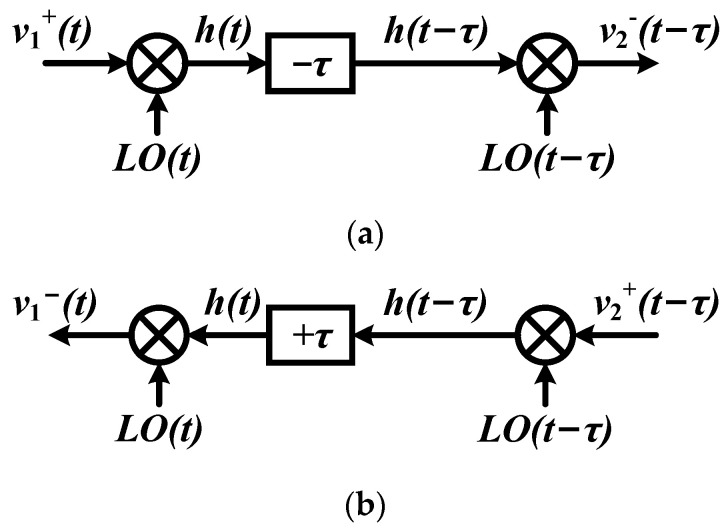
Doubly square-wave mixing system with time delay in the internal node: (**a**) forward direction; (**b**) reverse direction.

**Figure 4 micromachines-14-00033-f004:**
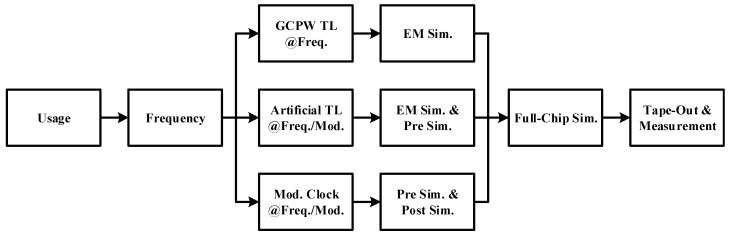
Comprehensive research framework of proposed circulator.

**Figure 5 micromachines-14-00033-f005:**
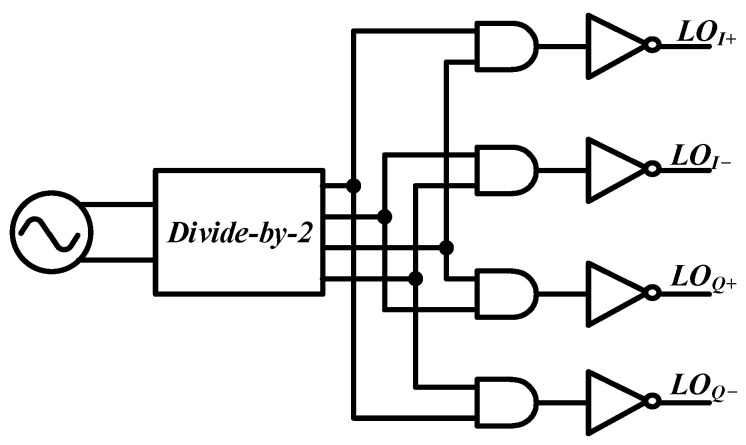
The 25% duty-cycle clock generation using a frequency divider and AND gates.

**Figure 6 micromachines-14-00033-f006:**
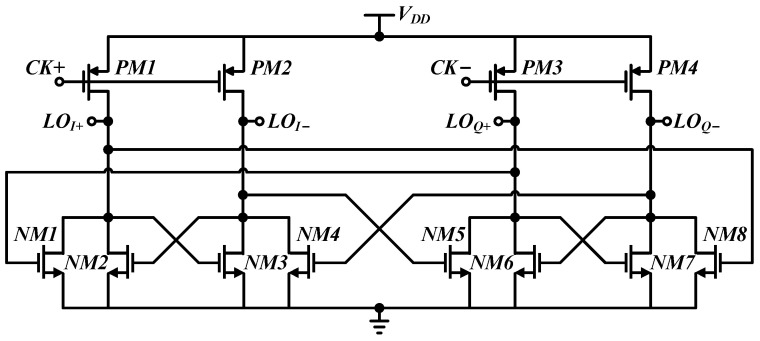
The 25% duty-cycle clock generation directly after the divide-by-2 frequency divider.

**Figure 7 micromachines-14-00033-f007:**
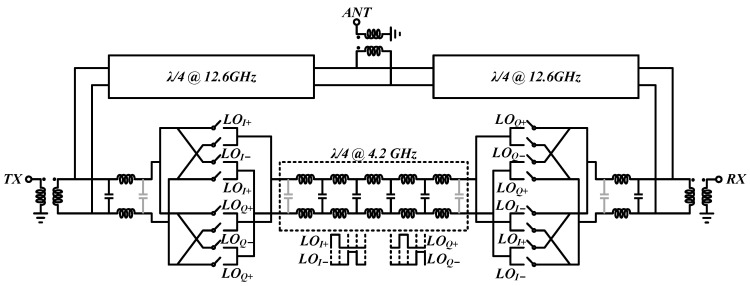
Proposed circulator using 25% duty-cycle clock signal.

**Figure 8 micromachines-14-00033-f008:**
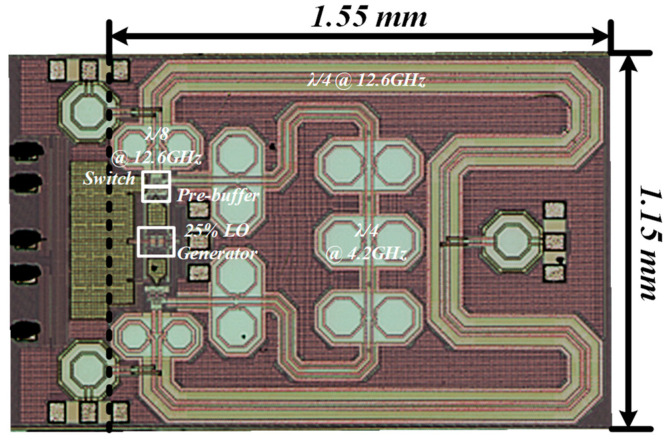
Die photo.

**Figure 9 micromachines-14-00033-f009:**
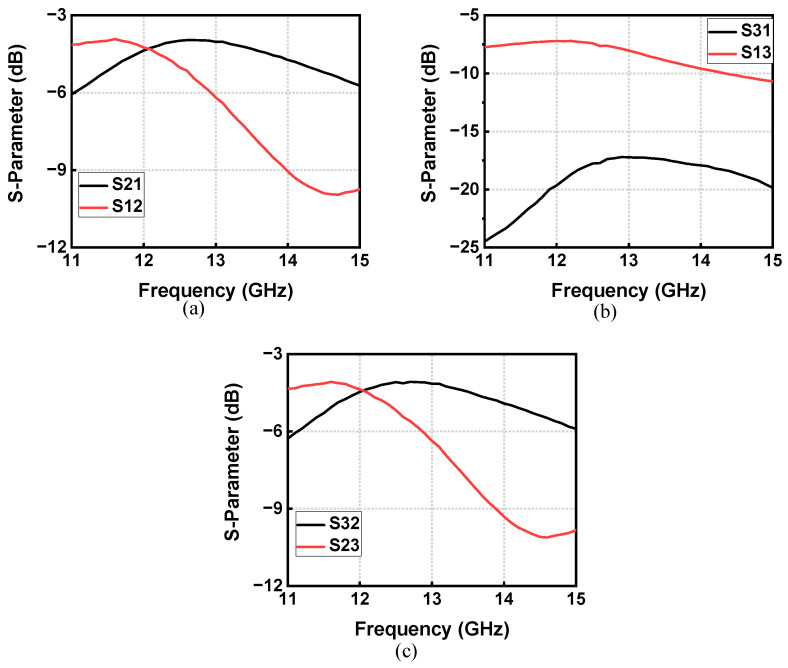
S-parameter measurements: (**a**) *S*_21_ and *S*_12_; (**b**) *S*_31_ and *S*_13_; (**c**) *S*_32_ and *S*_23_.

**Figure 10 micromachines-14-00033-f010:**
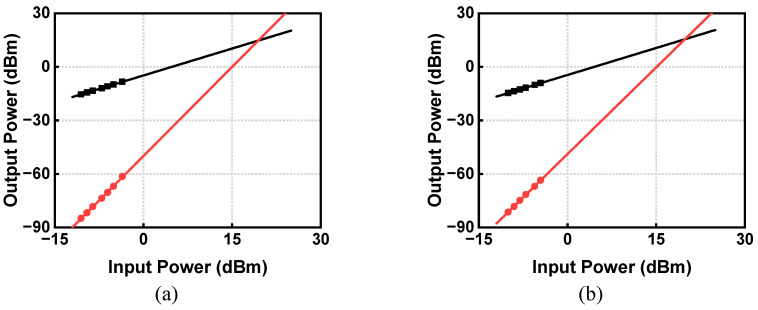
IIP3 measurements (**a**) TX to ANT; (**b**) ANT to RX.

**Table 1 micromachines-14-00033-t001:** Summary of prior literature and proposed work.

Reference	Topology	Technology	Frequency (GHz)	Clock Duty Cycle	Usage
[36]	N-path	65 nm	0.6–0.8	8	Sub-GHz
[40]		65 nm	0.11–1.1/ 0.28–1.15	4/8	Sub-GHz
[41]		65 nm	DC-1	8	Sub-GHz
[37] [42]	Switched TL	45 nm SOI * 180 nm SOI	22.7–27.3 0.86–1.08	2 2	5G Sub-GHz
[43]		45 nm SOI	50–56.8	2	WiFi
[44]		180 nm SOI	0.914–1.086	2	Sub-GHz
[45]		28 nm FDSOI **	0.05–7	2	Sub-GHz

* SOI: Silicon-on-insulator; ** FDSOI: Fully depleted silicon-on-insulator.

**Table 2 micromachines-14-00033-t002:** Phase shift in forward direction.

Modulation Period	Left	TL *	Right	Phase Shift
0	*A*_1_ (0°)	90°	*A*_2_ (0°)	90°
*T_m_*/4	*C*_1_ (0°)	90°	*C*_2_ (0°)	90°
*T_m_*/2	*B*_1_ (180°)	90°	*B*_2_ (180°)	450° (90°)
3*T_m_*/4	*D*_1_ (180°)	90°	*D*_2_ (180°)	450° (90°)

* TL = The λ/4 transmission line in the gyrator.

**Table 3 micromachines-14-00033-t003:** Phase shift in reverse direction.

Modulation Period	Right	TL *	Left	Phase Shift
0	*D*_2_ (180°)	90°	*C*_1_ (0°)	270° (−90°)
*T_m_*/4	*A*_2_ (0°)	90°	*B*_1_ (180°)	270° (−90°)
*T_m_*/2	*C*_2_ (0°)	90°	*D*_1_ (180°)	270° (−90°)
3*T_m_*/4	*B*_2_ (180°)	90°	*A*_1_ (0°)	270° (−90°)

* TL = The λ/4 transmission line in the gyrator.

**Table 4 micromachines-14-00033-t004:** Comparison table with the state-of-the-art.

	This Work	[37]	[42]	[43]	[47]	[23]
Type	Passive Circulator	Passive Circulator	Passive Circulator	Passive Circulator	Passive Circulator	Active Quasi-Circulator
Technology	90-nm CMOS	45-nm CMOS SOI	180-nm CMOS SOI	45-nm CMOS SOI	180-nm CMOS	180-nm CMOS
Frequency (GHz) *	11.6–14.3	22.7–27.3	0.86–1.08	50–56.8	0.89–0.92	1–7 ****
Designed Center Frequency (GHz)	12.6	25	0.95	54	0.91	N/A
Reconfiguration /Tuning	No	Clock Phase Tuning	Antenna Tuning	Clock Phase Tuning	External Reconfigurable Impedance Tuning	No
TX-ANT Transmission (dB)	−3.9 @12.6 GHz	−3.3 @25 GHz	−2.1 @0.95 GHz	−3.6(1.2V V_DD_)/−3.9(1V V_DD_) @54 GHz	−4.8 @0.91 GHz	>−10
ANT-RX Transmission (dB)	−4.0 @12.6 GHz	−3.2 @24.7 GHz	−2.9 @0.95 GHz	−3.1 @54 GHz	−4.8 @0.91 GHz	>−9
TX-RX Isolation (dB) **	> 17.2	> 18.3	> 25	> 20	> 20	>36
Isolation Bandwidth ***	21.4%	18.4%	17%	14.6%	3.4%	N/A
TX-ANT IIP3 (dBm)	19.7	20.1	50.025	19.43	6.1	9.7
ANT-RX IIP3 (dBm)	20	19.9	36.9	19.03	6	3.5
Area (mm^2^)	1.55 × 1.15	1.8 × 1.2	4.6 × 3.6	1.57 × 1.1	36	1.03 × 0.55

* Frequency range of 1-dB IL degradation; ** Isolation in the frequency range; *** Frequency range divided by designed center frequency; **** Secured isolation bandwidth.

## Data Availability

Not applicable.

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
