# Peer review of "Analysis and Design of a Non-Magnetic Bulk CMOS Passive Circulator Using 25% Duty-Cycle Clock"

_micromachines, 2022, doi:10.3390/mi14010033_

Round 1

Reviewer 1 Report

This article presents a non-magnetic CMOS circulator based on switched transmission line architecture with 25% duty cycle clocks:

·        Line 38: Did the authors mean the transistors biased in the active region suffer from noise? Most circulators implemented on CMOS use transistors even the implementation presented in this article.

·        Line 54: As shown in the analysis section, this architecture with 25% duty cycle clocks has not resulted in better insertion loss. So, the authors should correct this statement.  

·        Line 95: The proposed architecture is loss-free just like the one proposed with 50% duty cycle clocks. However, in this architecture with 25% duty cycle clocks, a total of 16 switches are clocked which is a factor of 2x higher than the 50% duty cycle clock architecture. This translates to 2x higher power consumption and also 2x higher parasitics. In a real implementation, I believe this architecture performs poorly and require a larger PDC compared to prior implementations.

o   So, why should one use this new proposed architecture?  Authors should clearly demonstrate the advantage of this proposed architecture over prior proposed architecture.  

·        Table 3: Contains errors in references. Add the 60GHz circulators as well to the comparison table.

·        Measurements: a) Noise figure, and b) P1dB measurements should be added. c) The overall spectral performance has to be shown to show that spurious harmonics are not generated.

·        The insertion losses of this implementation seem to be higher than in prior works. Authors should elaborate on the sources of insertion losses and how to mitigate them. 

Reviewer 2 Report

Paper is well written and formulated clearly but technically it needs some major revisions.

Abstract:

Follow the basic pattern of template of abstract formation….as:

We strongly encourage authors to use the following style of structured abstracts, but without headings: (1) Background: Place the question addressed in a broad context and highlight the purpose of the study; (2) Methods: Describe briefly the main methods or treatments applied; (3) Results: Summarize the article's main findings; and (4) Conclusions: Indicate the main conclusions or interpretations. (5) Add something about the benefits results of the research.  ……also give quantitative results in the abstract. 

[Please revise-according to point 2,3,4 and 5 Conclusion and benefits]

Introduction:

-Can you please establish the research gap [Please add]

-Please add a data table along with citation in the manuscript considering following points:

(a) Previously used method/leaves of other trees for same purpose (b) New ideeas/leaves of other trees for same purpose to be studied (c) Previously asked Questions (d) Previously used techniques for similar type of research.

-Establish your research questions in this section.

-Update your literature review with more additional references of published work in recent years.

-So, establish a research gap and connect your research methodology, data analysis, results with that research gap and produce a discussion on future directions.

Methods

-A Comprehensive research framework is missing (flowchart). This portion should be written in step wise pattern so that readers can understand the procedure for implementation purpose. [Please add]

-Also add references(citations) related to applied methodology.

Limitations of the study:

-Please add heading about the limitations of the study.

-Suggestion/recommendation to the policy makers must be there and it is responsibility of scientist to device such solution which are practically applicable.

Best regards, 

Round 2

Reviewer 1 Report

I don't think the switched transmission line gyrator with 25% clocks has a higher gain. Ideally, this architecture just like the 50% duty cycle architecture, just exhibits a gain of unity. 

Please explain the advantages of this proposed architecture over prior work clearly.
